# New Poly(Propylene Imine) Dendrimer Modified with Acridine and Its Cu(II) Complex: Synthesis, Characterization and Antimicrobial Activity

**DOI:** 10.3390/ma12183020

**Published:** 2019-09-18

**Authors:** Paula Bosch, Desislava Staneva, Evgenia Vasileva-Tonkova, Petar Grozdanov, Ivanka Nikolova, Rositsa Kukeva, Radostina Stoyanova, Ivo Grabchev

**Affiliations:** 1Institute of Science and Technology of Polymers, Institute of Science and Technology of Polymers-Spanish National Research Council (ICTP-CSIC), Juan de la Cierva 3, 28006 Madrid, Spain; 2Department of Chemical Technology, University of Chemical Technology and Metallurgy, 1756 Sofia, Bulgaria; grabcheva@mail.bg; 3The Stephan Angeloff Institute of Microbiology, Bulgarian Academy of Sciences, 1113 Sofia, Bulgaria; Evaston@yahoo.com (E.V.-T.); grozdanov_bg@yahoo.com (P.G.); vanianik@mail.bg (I.N.); 4Institute of General and Inorganic Chemistry, Bulgarian Academy of Sciences, 1113 Sofia, Bulgaria; rositsakukeva@yahoo.com (R.K.); radstoy@svr.igic.bas.bg (R.S.); 5Faculty of Medicine, Sofia University “St. Kliment Ohridski”, 1407 Sofia, Bulgaria

**Keywords:** dendrimer, metallodendrimer, acridine, antimicrobial activity, antibacterial cotton

## Abstract

A second-generation poly(propylene imine) dendrimer modified with acridine and its Cu(II) complex have been synthesized for the first time. It has been found that two copper ions form complexes with the nitrogen atoms of the dendrimeric core by coordinate bonds. The new compounds have been characterized by nuclear magnetic resonance (NMR), electron paramagnetic resonance (EPR), fourier-transform infrared spectroscopy (FTIR) and fluorescence spectroscopy. The spectral characteristics of the modified dendrimer have been measured in different organic solvents, and a negative fluorescence solvatochromism has been observed. The antimicrobial activity of the dendrimers has been tested against model pathogenic microorganisms in agar and by broth dilution method. The cotton fabric treated with both dendrimers has been evaluated towards pathogenic microorganisms. The obtained modified cotton fabrics have been shown to hamper bacterial growth and to prevent biofilm formation. Dendrimer cytotoxicity has been investigated in vitro in the model HEp-2 cell line.

## 1. Introduction

Antibacterial surfaces are very important with regard to minimizing infectious diseases which are one of the main causes of mortality worldwide [1]. This problem is mainly due to the increasing resistance of pathogenic microorganisms to antibiotics applied in clinical practice [2] Heterocyclic compounds have major role in the design and investigations of new bioactive drugs [3,4] Therefore acridine derivatives are one of the intensively exploited organic fluorophores in which fluorescence color intensity depends strongly on the polarization of their chromophoric system [5,6]. Possessing a heteroaromatic polycyclic molecule acridine derivatives are well known for their DNA intercalating abilities and pharmacological activity. That has led to the design and preparation of acridine compounds with anticancer, antimalarial, antiviral and antifungal activities [5,6,7,8,9,10].

Dendrimers are macromolecules with well-defined molecular weight and a high degree of branching units containing different reactive functional groups to which substances with biological activity may be attached by a chemical bond or by weak intermolecular interactions [11]. Compared to the low molecular weight bioactive compounds, dendrimers have the potential to deliver a large dose of biologically active substances, depending on the dendrimer generation. [12,13]. On the other hand, dendrimers containing metal ions, or metal nanoparticles also exhibit good biological activity [14,15,16,17].

Cotton fabrics are natural polymers, that have properties, such as high water absorptivity and air permeability and which are used for the production of textile products with different applications [18,19]. One of the major drawbacks of ordinary cotton fabrics is the possibility of being colonized by pathogenic microorganisms forming biofilms leading to undesired hygienic problems [20,21]. In the medical and clinical practice, cotton materials are used to obtain antibacterial textiles like wound dressings, hospital linen, sterile surgical materials, etc. [22]. Therefore, the improvement of their antimicrobial resistance is of major importance.

In this paper, the peripheral modification of a second-generation poly (propylene imine) dendrimer with acridine fluorophores has been described for the first time. Its copper complex has also been obtained and characterized. The spectral characteristics of the new fluorescent dendrimer have been investigated in different organic solvents. The antimicrobial activity of the dendrimers in agar, liquid medium and after their deposition on a textile material has also been tested. The cytotoxicity of the dendrimers has been evaluated in vitro in the model HEp-2 cell line.

## 2. Materials and Methods

### 2.1. Materials

Poly(propylene imine) dendrimer second-generation, phenol, 9-chloroacridine, ammonia solution (32%) and anhydrous magnesium sulfate were obtained from Sigma-Aldrich and used as received. All organic solvents: tetrahydrofurane, ethylacetat, chloroform, dichloromethane, acetonitrile, methanol, ethanol, used in this study were used as obtained from Sigma-Aldrich without any additional purifications. Dimethyl sulfoxide (DMSO) for molecular biology was used for antibacterial screening (Sigma–Aldrich). The antibacterial activity of the compounds was tested in vitro with Gram (+) bacterial strain *Bacillus cereus* (*B. cereus*) ATCC 11778 and Gram (−) bacterial strain *Pseudomonas aeruginosa* (*P. aeruginosa*) 1390, and the antifungal activity was screened on fungi strain *Candida lipolytica* (*C. lipolytica*) 7618. Microorganisms were obtained from the collection of the Institute of Microbiology, Bulgarian Academy of Sciences, Sofia, Bulgaria. Before experimental use, cultures from solid medium were sub cultivated in a liquid medium (meat-peptone broth, MPB), incubated (orbital shaker incubator) and used as a source of inoculums for each experiment.

### 2.2. Synthesis of Acridine Dendrimer (ACR)

A two-neckle round bottom flask, provided with a reflux cooler, magnetic stirring and N_2_ atmosphere was loaded with 0.5 g of 9-chloroacridine (2.3 mmol) and 1.15 g of distilled phenol. The reaction was maintained refluxing at 110 °C for 1 h. Then, 182.8 L of dendrimer (0.181 g, 0.23 mmol) was added, and reflux at 110 °C was maintained for 6 h. The reaction was then allowed to cool at room temperature, and 30 mL of acetone were added, under stirring, to the brown slurry obtained. The slurry solidified, and the yellow-orange solid obtained was washed with acetone. The solid was dissolved in water/NH_4_OH (pH 8) and extracted with dichloromethane (4 × 20 mL). The organic extracts were dried over anhydrous MgSO_4_ and evaporated under vacuum. 0.25 g (yield = 52%) of a yellow-orange solid were obtained. Tdecomp: 254 °C

FT-IR cm^−1^: 3240, 1615, 1594, 1558, 1518, 1466, 1138, 1020, 752, 647.

^1^H-RMN (CDCl_3_) ppm: 1.22 (s, 4H^15^, C-CH_2_-CH_2_-C), 1.57 (bs, 16H^9^ and 8H^12^, C-CH_2_-C ), 1.91–2.07 (bs, 8H^11^, 8H^13^ and 4H^14^, N-CH_2_), 2.28 (bs, 16H^10^, N-CH_2_), 3.60 (bs, 16H^8^, NH-CH_2_-), 7.03 (bs, 16H^2^, Ar-H), 7.43 (bs, 16H^3^, Ar-H), 7.89 (sa, 16H^4^ and 16H^1^, Ar-H).

^13^C-NMR (CDCl_3_) ppm: 23, 25, 29, 45, 51, 55, 115, 117, 121, 126, 129, 133, 141, 152.

API-ES-MS (positive) *m*/*z*: found: 2191.6 (100, [M + H]^+^), 1096.1 (65, [M + H]^+^/2).

### 2.3. Synthesis of [Cu_2_(ACR)(NO_3_)_2_] 

ACR dendrimer (0.219 g, 0.1 mmol) was dissolved in 20 mL of ethanol and Cu(NO_3_)_2_·3H_2_O (0.113 g, 0.6 mmol) was added to the solution. The mixture was stirred for 2 h, and the solid formed was filtered, washed with ethanol three times, and dried under vacuum. Yield: 0.225 g, 88.1%

FT-IR cm^−1^: 1638, 1691, 1531, 1466, 1367, 1298, 1173, 1037, 826, 754, 661.

Analysis: C_144_H_152_N_26_O_12_Cu_2_ (2564.8 g mol^−1^) Calculated (%): C 67.43, H 5.93, N 14.20; Found (%): C 67.54, H 5.88, N 14.28

### 2.4. Dendrimers Characterization

The UV-Vis spectrophotometric investigations of the dendrimers were performed on a UV-Vis “Thermo Spectronic Unicam UV 500” double beam spectrophotometer. Fluorescence spectra were taken on a “Cary Eclipse” fluorimeter. The absorption and fluorescence spectra were recorded using 10^−6^ mol/L solutions of the dendrimers. The quantum yield of fluorescence was determined by comparing the areas underneath the fluorescence spectra of the dendrimers. Standard quinine bisulfate/H_2_SO_4_ 1N (Φ_f_ = 0.546) was used as standard material to calculate the dendrimer fluorescence quantum yields. ATR FT-IR spectroscopic analyses of dendrimers were performed using an IRAffinity-1 spectrophotometer (Shimadzu Co., Kyoto, Japan) equipped with a MIRacleTM ATR (diamond crystal, depth of penetration of the infrared (IR) beam into the sample was about 2 mm) accessory (PIKE Technologies, Cottonwood, WI, USA). The spectra were recorded from 4000 cm^−1^ to 500 cm^−1^ with a spectral resolution of 4 cm^−1^ using a DLATGS detector equipped with a temperature controller. All spectra were corrected for H_2_O and CO_2_ using IR solution internal software. ^1^H (600.13 MHz) and ^13^C (150.92 MHz) NMR spectra were acquired on an BRUKER, AVANCE AV600 II+NMR spectrometer (Rheinstetten, Germany). The measurements were carried out in a CDCl_3_ solution at ambient temperature. The chemical shifts were referenced to a tetramethylsilane (TMS) standard. The EPR spectra of Cu(II) complexes were recorded as the first derivative of the absorption signal by using a Bruker EMXplus EPR spectrometer (Rheinstetten, Germany), operating in the X-band (9.4 GHz). The recording temperature was varied within the range of 120–450 K. The quantitative EPR calculations were performed by SpinCountTM software module (Bruker, Hamburg, Germany). The spectra were simulated using the program SIMFONIA (Bruker, Hamburg, Germany). The effect of the metal cations upon the fluorescence intensity was examined by adding a few L of stock solution of the metal cations to a known volume of the tripod solution (3 mL). The addition was limited to 0.1 mL, so that dilution remained insignificant.

### 2.5. Cotton Fabric Functionalization with ACR and [Cu_2_(ACR)(NO_3_)_2_]

0.005 g of each dendrimer was dissolved in 5 mL of a N,N-dimethylformamide(DMF)–water 1:4 (*v*/*v*) solution. The cotton fabric sample (1 g) (weight 140 g m^−2^) was immersed into the solution at 25 °C for 30 min, washed with water and dried at ambient temperature. The dyed cotton fabric is yellow in color and resistant to water treatment. In this case, the fixation of the dendrimer on the surface of the textile material is accomplished by van der Waals forces and possible hydrogen bonds between hydroxyl groups from the cellulose structure and tertiary nitrogen atoms from the dendrimer molecule.

### 2.6. Cellular Toxicity

HEp-2 cells (National Bank for Industrial Microorganisms and Cell Cultures, No. NBIMCC-95, Sofia, Bulgaria) were grown in medium containing 10% heated calf serum in DMEM (Gibco BRL, Red Bank, NJ, USA) supplemented with 10 mmol/L HEPES buffer (Gibco BRL, Red Bank, NJ, USA) and antibiotics (penicillin, 100 U/mL; streptomycin, 100 µg/mL). 

Monolayer cells in 96-well plates (Costar®, Corning Inc., Kennebunk, ME, USA) were inoculated with 0.1 mL/well-containing concentrations (in logarithmic intervals) of the compounds diluted in a maintenance medium. Cells were incubated in a humidified atmosphere at 37 °C and 5% CO_2_ for 48 h. After microscopic evaluation, the maintenance medium containing the test compound was removed, cells were washed, and 0.1 mL maintenance medium supplemented with 0.005% neutral red dye was added to each well and cells were incubated at 37 °C for 3 h. After incubation, the neutral red day was removed, and cells were washed once with PBS, and 0.15 mL/well desorb solution (1% glacial acetic acid and 49% ethanol in distilled water) was added. The optical density (OD) of each well was read at 540 nm in a microplate reader (Organon Teknika Reader model 530, Organon, West Chester, PA, USA). The 50% cytotoxic concentration (CC_50_) was defined as the material concentration that reduced the cell viability by 50% when compared to untreated control.

### 2.7. Antimicrobial Activity Test

The antimicrobial activity of the compounds was firstly tested by the agar diffusion method using 0.2% solutions of the investigated compounds in dimethyl sulfoxide (DMSO). Plates containing Mueller-Hinton agar (MHA) were inoculated with aliquots of suspensions of microbial cultures. An equal amount (30 µL, 0.06 µg) of each sample solution was introduced into wells (8 mm in diameter) punched in MHA following a sterile procedure. Standard commercial discs with gentamicin (G, antibacterial agent) and nystatin (Ns, antifungal agent) were used as references. A positive control using only inoculation and negative control using only DMSO in wells were also prepared. The plates were incubated at the appropriate temperature for 24−48 h, and the resulting inhibition zones (diameter, mm) were recorded.

Broth dilution method was performed for in vitro determination of the minimum inhibitory concentration (MIC) of the compounds [23]. Stock solutions of the investigated compounds (0.1% in DMSO) were serially diluted in MPB to final concentrations ranging from 2 to 200 µg/mL. After inoculation, the test tubes were incubated at the appropriate temperature for 24 h under shaking. Positive controls (compounds and MPB, without inoculum) and negative controls (MPB and inoculum, without compounds) were also prepared. Growth of the strains was assayed by monitoring the turbidity at 600 nm (OD_600_). Microbial growth (%) was determined on the basis of the positive control, which was considered as 100%. The MIC was considered to be the lowest concentration of the tested sample to inhibit the visible growth of microorganisms. All assays were performed in triplicate, and the average was taken; standard deviations were less than 5%.

### 2.8. Antibacterial Activity of Cotton Fabrics

The antibacterial activity of cotton fabrics treated with ACR and [Cu_2_(ACR)(NO_3_)_2_] was tested against *B. cereus* and *P. aeruginosa* as model strains. Test tubes containing MPB and square cotton specimens (10 mm × 10 mm) were inoculated with a suspension of each bacterial culture. Untreated cotton and without specimens were used as controls. After incubation for 24 h at the appropriate temperature, the bacterial growth was determined by measuring OD_600_. Antimicrobial activity of the treated cotton samples was evaluated by the reduction of OD_600_ after incubation compared to the control sample. All antimicrobial tests were done in triplicate, and the average was taken.

### 2.9. Scanning Electron Microscopy (SEM)

The adhesion and biofilm formation over the cotton fabric were assessed by SEM. The tubes containing MPB and specimens of untreated and treated cotton fabrics were inoculated with a suspension of *P. aeruginosa.* After incubation for 24 h, the specimens were washed with phosphate-buffered saline, dried and coated with gold with Jeol JFC-1200 fine coater (Jeol Ltd., Tokyo, Japan) and then investigated by Jeol JSM-5510 SEM (Jeol Ltd., Tokyo, Japan).

### 2.10. Hydrophilicity of Cotton Fabrics

Static immersion test was carried out to measure the amount of water absorbed by the cotton fabrics. The tested cotton samples were weighed and immersed into distilled water, taken out after about 3 min and tapped to remove excess water and then weighed once again. The absorption percentage was determined by the following formula [24]:
Absorption (%) = (mass of water absorbed/original mass) × 100

## 3. Results

### 3.1. Synthesis of Acridine Dendrimer

To obtain a polypropylene imine dendrimer modified with acridine fluorophores, we used a second-generation dendrimer containing eight primary amino groups in its periphery, thus enabling eight acridine fragments to be coupled into one molecule. Fully functionalized acridine dendrimer (ACR) was obtained in a moderate yield by modification of a previously described procedure, [25] as depicted in Scheme 1. As the acridine dendrimer was obtained as a protonated species in the reaction medium, the solid was dissolved in water/NH_4_OH (pH 8) wherefrom it was extracted in its neutral form. The copper complex, [Cu_2_(ACR)(NO_3_)_2_], was subsequently synthesized easily by a reaction run in ethanol at room temperature.

### 3.2. Spectral Characterizations

The dendrimer is insoluble in water, but it is very well soluble in organic solvents. The basic spectral characteristics of ACR dendrimer: Absorption (λ_A_) and fluorescence (λ_F_) maxima, Stoke’s shift ν_A_−ν_F_), the quantum yield of fluorescence (Φ_F_) have been investigated in seven organic solvents of different polarity, and the main results are summarized in Table 1.

The spectrum of the acridine dendrimer has absorption maxima between 396–412 nm, ascribed to an internal charge transfer (ICT) transition. It emits fluorescence with maxima at 455–480 nm region. In most solvents, the spectra are composed by a single broadband, whereas, in hydrogen bonding solvents (alcohols) a shoulder can be distinguished [26,27,28,29]. The results show solvent polarity dependence. Comparing the absorption maxima of the spectra taken in non-polar solvents with those taken polar ones, a positive solvatochromism has been obtained, while the respective maxima in polar environments do not change their position. In the case of fluorescence maxima, it is seen that with increasing the medium polarity, their values decrease and a negative solvatochromism has been found (Figure 1). The lower values of Stokes shift (ν_A_–ν_F_) in a polar medium can be explained by the dipole-dipole interactions and possibility of the formation of hydrogen bonds. That probably stabilizes the acridine molecules in the excited state, and conformational changes are slightly pronounced. This is also confirmed by the results obtained for the quantum fluorescence yield, where the values in polar media are several times higher (Table 1). Similar results have been obtained when acridine functionality has been bonded to a hyperbranched polymer [30].

In order to elucidate the formation of a copper complex, titration of ACR dendrimer with Cu(II) ions has been carried out in acetonitrile solution. Figure 2A shows the decrease of fluorescence intensity with increasing concentration of Cu(II) ions. Figure 2A also shows that Cu(II) forms a complex with a dendrimer molecule at a 1:2 stoichiometry. The possible formation of the coordination of Cu(II) is with the tertiary amino groups in the dendrimer core. IR spectroscopy has been used for the characterization of the isolated solid complex (Figure 2B). The difference is observed in the range 1200–1340 cm^−1^, where is the absorption of nitrate groups (−NO_3_). 

### 3.3. EPR Analysis

Figure 3 display the EPR spectra of the dendrimer complexes [Cu_2_(ACR)(NO_3_)_2_ and EPR spectrum of ACR ligand is shown for the comparison, The EPR spectrum of the copper complex consists of an anisotropic signal with *g*-components of *g*_1_ = 2.238, *g*_2_ = 2.075, *g*_3_ = 2.065 between 120 and 295 K. The *g*-components are not sensitive towards the recording temperature. In the range of g_1_-component, a hyperfine structure is hardly resolved, the constant hyperfine being A_1_ = 16.5 mT. The analysis of EPR parameters of the Cu(II) complex reveals that the *g_1_*-component adopts a relatively low value, whereas, the magnitude of the hyperfine constant A_1_ is relatively high. On the basis of Peisach-Blumberg diagram [31], the relation between the values of *g*- and *A*-components can be used as an experimental measure on the composition of the coordination shell around Cu(II) ions: The coordination of nitrogen to Cu(II) provokes a decrease of the g_||_-value and a corresponding increase in A_||_ value, while the opposite trend is observed when oxygen is coordinated around Cu(II). The comparison shows that for the complexes of Cu(II) with ACR, the ligand is coordinated to Cu(II) ions mainly through nitrogen atoms (Scheme 1).

The ligand displays a narrow symmetrical signal with a *g*-factor of 2.003, in comparison with the copper complex. The *g*-value and the extremely narrow line width (less than 0.5 mT) implies that the signal comes, most probably, from free radicals. It should be taken into account that the origin of this signal is unclear. It is of importance that the signal, due to the ligand is not observed in the EPR spectrum of the copper complex; hence, all ligand molecules are involved in the complexation.

### 3.4. NMR Characterization of Dendrimer ACR

The chemical structure of the dendrimer ACR was confirmed by ^1^H and ^13^C NMR spectroscopy. The ^1^H NMR data show the evidence for the bonding of acridine fluorophore to the primary amino group from the dendrimer structure. From Scheme 1 it is clearly seen that the chemical structure of the polypropylene imine dendrimer is composed only of methylene groups (−CH_2_−) and their characteristic signals are in the spectral range δ = 1.22–3.60 ppm. These signals were observed at polypropylene imine dendrimers modified with 1,8-naphthalimide or benzanthrone fluorophores [32,33]. The aromatic proton (Ar−H) signals from the acridine structure are recorded in the δ = 7.03–7.89 ppm region as broad singlets. The carbon signals for acridine ring (Ar−C) are at δ = 115–152 ppm and that of the methylene groups −CH_2_− group of the poly(propylene imine) dendrimer framework are between δ = 23–55 ppm.

### 3.5. Antimicrobial Activity of ACR and [Cu_2_(ACR)(NO_3_)_2_]

Figure 4 plots the inhibition zones formed by the ACR and [Cu_2_(ACR)(NO_3_)_2_]) against three different model pathogens. Both the free ligand and [Cu_2_(ACR)(NO_3_)_2_] showed strong activity against the used Gram (+) *B. cereus* and the yeasts *Candida*; the complex exhibited about 30% higher activity than the ligand and standard drug. The low activity has been observed against Gram negative *P. aeruginosa*. 

The MICs of the compounds have been determined by serial dilution in MPB, and the results are shown in Table 2. Obviously, complexation with Cu(II) ions enhances the antimicrobial effect of the ligand to a different extent, depending on the tested strain. 

MIC s of the [Cu_2_(ACR)(NO_3_)_2_] vary from 10 to 80 µg/mL, and from 40 to 120 µg/mL for the ligand. Both the ligand and its Cu(II) complex exhibit the highest effectiveness in inhibiting the growth of *B. cereus* (the lowest MICs 10 and 40 µg/mL, respectively) followed by *C. lipolytica* with MICs of 40 and 80 µg/mL, respectively. Gram (−) *P. aeruginosa* displays the highest resistance to the compounds with the highest MIC values. The observed difference in the susceptibility of both type of bacteria to the investigated substances is due to the different structure of the bacterial cell walls. [34].

Dendrimer interior is a binding site for Cu(II) ions, and after its complexation, the polarity of the Cu(II) is reduced which can be explained to the partial sharing of positive charges with the nitrogen groups that increase the lipophilicity of the [Cu_2_(ACR)(NO_3_)_2_] complex. The enhanced lipophilicity increases penetration of the metal complex through the lipid membranes and consequently blocking metal-binding sites in the targeted enzymes of the microorganisms [35]. 

According to Holetz theory MICs values can be used to classify the microbiological activity of the compounds against pathogenic microorganisms [36]. Compounds with MICs less than 100 µg/mL exhibit good antimicrobial activity, in the range 100 to 500 µg/mL have moderate activity, weak activity demonstrate compounds in the range from 500 to 1000 µg/mL, and compounds over 1000 µg/mL are considered inactive. Depending on those criteria, our results exhibit a good antimicrobial activity of the tested compounds against the used model strains.

### 3.6. Antibacterial Activity of Cotton Fabric

Various methods have been proposed for the antimicrobial modification of material surfaces as an alternative way of preventing the formation of highly resistant biofilms [37]. The antimicrobial activity of cotton fabrics treated with the compounds has been investigated by reduction of the growth of Gram (+) *B. cereus* and Gram (−) *P. aeruginosa* used as model strains. The results demonstrate good antimicrobial effect of both ACR and [Cu_2_(ACR)(NO_3_)_2_] treated cotton fabrics (Figure 5). They are highly effective inhibiting about 90% of the growth of *B. cereus* and more than 50% of the growth of *P. aeruginosa*. Slow diffusion of the compounds from the cotton fabric into the medium and the direct contact with bacterial cells may contribute to the antimicrobial effect of the modified cotton fabrics.

### 3.7. Hydrophilicity of Cotton Fabrics

Hydrophilicity of cotton fabrics has been assessed by the amount of absorbed water. Hydrophilicity of untreated cotton fabric and cotton fabrics treated with ACR and [Cu_2_(ACR)(NO_3_)_2_] complex has been determined to be 112%, 95% and 86%, respectively. The results clearly indicated that the compounds induce an increase in hydrophobicity when applied to cotton fabric may be due to an increase in the number of hydrophobic groups on the surface of the treated cotton fabric. More effective hydrophobicity has been achieved with the cotton fabric treated with [Cu_2_(ACR)(NO_3_)_2_], if compared to that of ACR treated fabric, which effect can be explained by the enhanced lipophilicity of the [Cu_2_(ACR)(NO_3_)_2_] complex.

### 3.8. SEM Analysis

*Pseudomonas aeruginosa* is an excellent biofilm producer, and it has been selected to investigate the efficacy of the cotton fabrics treated with the ACR and [Cu_2_(ACR)(NO_3_)_2_] in preventing the adhesion and biofilm formation [38]. In Figure 6 are presented SEM micrographs of untreated cotton fabrics and cotton fabrics treated with ACR and its [Cu_2_(ACR)(NO_3_)_2_] complex after 24 h of incubation. From the SEM images, it is clearly seen that a formation of P. aeruginosa biofilm on the untreated cotton fabric (Figure 6B). In this case, many bacterial cells are adhered to the cotton surface and embedded into extracellular bio matrix. On the other hand, а significant reduction of biofilm formation and bacterial the adhesion has been observed on the surface of the cotton fabric treated with ACR (Figure 6D). Only single bacterial cells are attached to the cotton surface of the cotton fabric treated with the complex [Cu_2_(ACR)(NO_3_)_2_] (Figure 6C). Thus, the deposition of ACR and [Cu_2_(ACR)(NO_3_)_2_] complex on the cotton fabric prevents the formation and proliferation of a bacterial biofilm, allowing the production of antibacterial cotton fabric.

### 3.9. In Vitro Cytotoxicity Assay

The evaluation of cytotoxicity of antimicrobials is a critical step to guarantee their safe use. The results for the cytotoxicity of the newly synthesized dendrimers are shown in Figure 7. It has been found that both acridine dendrimer and its Cu-complex affected HEp-2 cells at similar doses, with CC_50_ values of 19.5 µg/mL and 19.4 µg/mL, respectively. At the active concentration of the [Cu_2_(ACR)(NO_3_)_2_] against Gram-positive *B. cereus* (MIC 10 µg/mL) cell viability is near 75%, while other tests did not detect viable cells at the active concentrations of the dendrimers (MICs 40–120 µg/mL). The results also show that the binding of copper ions to the dendrimer molecule does not additionally increase the cytotoxicity of the dendrimer ligand ACR.

## 4. Conclusions

For the first time, acridine has been used for the modification of a poly(propylene imine) dendrimer (ACR). The new fluorescent dendrimer has been used as a ligand to obtained its Cu(II) complex [Cu_2_(ACR)(NO_3_)_2_]. EPR, FTIR, and fluorescent spectroscopy have been used to confirm the structure of metallodendrimer. It has been found that two copper ions are included in the dendrimer core by coordination with the inner tertiary nitrogen atoms. The basic photophysical characteristics of dendrimer have been examined in different organic solvents, and a negative fluorescence solvatochromism has been observed. The antimicrobial activity of dendrimers has been tested in vitro against some model Gram positive and Gram negative bacteria and yeasts. The results demonstrated enhancement in the antimicrobial activity of acridine dendrimer via complexation with copper ions against *B. cereus* and *C. lipolytica*. Deposition of dendrimers on the surface of cotton fabric has led to an increase in hydrophobicity of the textile. That prevents the formation of bacterial biofilm and makes these compounds useful for the production of antibacterial cotton fabrics.

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
