# Peer review of "New Poly(Propylene Imine) Dendrimer Modified with Acridine and Its Cu(II) Complex: Synthesis, Characterization and Antimicrobial Activity"

_materials, 2019, doi:10.3390/ma12183020_

Round 1

Reviewer 1 Report

In this manuscript, a second generation poly(propylene imine) dendrimer modified with acridine was synthesized and complexed with Cu(II). The chemical structures and the photophysical properties of the compounds were characterized. Then antimicrobial activities of the molecules were evaluated and they showed good antibacterial properties. The compounds treated cotton fabric inhibited bacterial growth and biofilm formation. Moreover, the cytotoxicities of the molecules were evaluated with model cells. Overall, this manuscript is of good quality and will benefit other researchers in this field after publication. But there are also some questions and problems.

1.        “1H-RMN” should be changed to “1H-NMR”

2.       It seems that Mass spectrometry is used for the characterization of the molecules “EM: m/z 2191.6”, but what does EM stand for?

3.       What is the water solubility of these molecules? Are they suitable for the cotton fabric applications? Will they stay in the fabric after sweating or washing?

4.       In Figure 5, it would be better if the authors could add some common anti-bacterial molecules for comparison.

5.       In Figure 7, the authors had better mark the biofilm in the pictures for the readers to understand.

6.       In Figure 8, at the active concentration of the compounds, significant cytotoxicity is observed and no viable cells could survive in many other occasions. But for the application in fabric, cells don’t meet the compounds in the solution directly. Could the authors provide some other experimental design to mimick the application better?

Author Response

1H-RMN” should be changed to “1H-NMR”

Corrected

It seems that Mass spectrometry is used for the characterization of the molecules “EM: m/z 2191.6”, but what does EM stand for?

The symbol has been corrected in the text.

What is the water solubility of these molecules? Are they suitable for the cotton fabric applications? Will they stay in the fabric after sweating or washing?

Explained in the text

In Figure 5, it would be better if the authors could add some common anti-bacterial molecules for comparison.

We have excluded Fig. 5 since zones of inhibition by the investigated compounds in comparison with the used standards are shown in Fig.4.

In Figure 7, the authors had better mark the biofilm in the pictures for the readers to understand.

We have changed the microphotograph of Figure 6B with a new one that better illustrates the presence of biofilm. The figure shows that all the cotton fiber is enveloped in bacteria and biofilm. While in Figure 6C biofilms are barely noticeable.

In Figure 8, at the active concentration of the compounds, significant cytotoxicity is observed and no viable cells could survive in many other occasions. But for the application in fabric, cells don’t meet the compounds in the solution directly. Could the authors provide some other experimental design to mimick the application better?

Unfortunately, we cannot provide data for cytotoxicity of the modified cotton fabrics

Reviewer 2 Report

The authors have reported a second generation of poly(propylene imine) dendrimer by chemical modification with acridine. The resulting compound was further modified via complexation with Cu(II), which have also been synthesized for the first time. The antibacterial activity of these new materials has been well proved.

As a first question: why the authors are saying “a second generation of poly(propylene imine) dendrimer” is there any first one ? than if yes please explain and highlight in the main bodytext (in the introduction for example)

The compounds have been characterized by NMR, EPR, FTIR and fluorescence spectroscopy. I would like to see a clear NMR data analysis (1H and 13C spectra, 2D HSQC and HMBC correlations and interpretation) as the compound is new. Please give 1H NMR of the starting material superimposed on that of the final product in order to clarify the aromatic substitution. Furthermore, HMBC correlations may help to confirm that the acridine is attached to the dendrimer main skeleton. Please use a separate supporting information document in order to report a full NMR characterization.

In the conclusion, the authors mentioned that there is an enhanced antimicrobial activity of acridine dendrimer complexe with copper ions against B. cereus and C. lipolytica. Is there any relationship with the counter ion NO3 ?

In my opinion, the article can be recommend for publication just after a fine NMR characterization showing all 1D and 2D spectra that may really help readership in understanding the utility of NMR as well as other techniques EPR, FTIR among others.

Author Response

As a first question: why the authors are saying “a second generation of poly(propylene imine) dendrimer” is there any first one ? than if yes please explain and highlight in the main bodytext (in the introduction for example)

Dendrimer generations start from the first and grow upwards. Depending on the generation, peripheral functional groups that can be modified are also increasing. Purva generation contains 4 amino groups, a second generation-8 amino groups, a third generation 16 amino groups, etc.
The choice of a second-generation dendrimer has been explained in the text.

The compounds have been characterized by NMR, EPR, FTIR and fluorescence spectroscopy. I would like to see a clear NMR data analysis (1H and 13C spectra, 2D HSQC and HMBC correlations and interpretation) as the compound is new. Please give 1H NMR of the starting material superimposed on that of the final product in order to clarify the aromatic substitution. Furthermore, HMBC correlations may help to confirm that the acridine is attached to the dendrimer main skeleton. Please use a separate supporting information document in order to report a full NMR characterization.

The chemical structure of the modified dendrimer is characterized by NMR (1H and 13C) spectra, FTIR and electron spectroscopy (absorption and fluorescence), API-ES-MS and elemental analysis. These are the methods that we think give a good characterization and purity of dendrimer compounds and that we have used so far in our practice. Due to the presence of paramagnetic copper in the metallodendrimer, EPR analysis was used to its characterisation.

In the conclusion, the authors mentioned that there is an enhanced antimicrobial activity of acridine dendrimer complexe with copper ions against B. cereus and C. lipolytica. Is there any relationship with the counter ion NO3?

In our studies for the preparation of metal complexes, we use nitrate salts of the corresponding metals, most commonly copper and zinc salts. In this work, we have used copper nitrate and the purpose of the study was to compare the properties of both dendrimers, a ligand (ACR) and the corresponding metallodendimer. To study the impact of counterions, we should use different copper salts.

Reviewer 3 Report

The paper demonstrates a mix of dendritic metal complexes and materials which benefit from their application. The metal complexes have been shown to demonstrate some antibacterial activity with differences observed between the metal complexed dendrimer and the dendrimer alone being somewhat similar.

The article is interesting to your readers however, before publishing the article requires amendments to address the somewhat disjointed display of experimentation. The introduction is acceptably written but does not flow through to the crux of the paper. Moreover, some of the sentences are nonsensical and grammatically incorrect.

This lack of clarity and structure continues into the results as the sections could be linked more clearly and the use of diagrams to help the reader be displayed better. For example, Scheme 1 and 2 could be combined as they do not need to be separated.

Nevertheless, I find the results somewhat disappointing as I do not understand why the dendrimer needs to be complexed?? Results show that the dendrimer alone is just as effective, if not more effective than the complexed version on the disc diffusion assays but in the MIC testing this is reversed? Whilst I agree with the assessment of why this is possible (Pg11 line 310 onwards) I would imagine that the dis diffusion assay is a more accurate pre-screen for how the molecules would act when coated on a fabric. Indeed, this appears to be the case as when the fabric is tested with both the dendrimer alone (ACR) and the complexed ACR the antibacterial results are virtually identical (similar to the disc diffusion assay). In addition, the cytotoxicity testing displays similar readings between the dendrimer alone and the complexed dendrimer meaning that I still am unable to understand why the complexed system is presented.

In conclusion, at this time I would reject this article.

Author Response

The article is interesting to your readers however, before publishing the article requires amendments to address the somewhat disjointed display of experimentation. The introduction is acceptably written but does not flow through to the crux of the paper. Moreover, some of the sentences are nonsensical and grammatically incorrect.

The English has been improved.

This lack of clarity and structure continues into the results as the sections could be linked more clearly and the use of diagrams to help the reader be displayed better. For example, Scheme 1 and 2 could be combined as they do not need to be separated.

Scheme 1 and 2 have been combined.

Nevertheless, I find the results somewhat disappointing as I do not understand why the dendrimer needs to be complexed?? Results show that the dendrimer alone is just as effective, if not more effective than the complexed version on the disc diffusion assays but in the MIC testing this is reversed? Whilst I agree with the assessment of why this is possible (Pg11 line 310 onwards) I would imagine that the dis diffusion assay is a more accurate pre-screen for how the molecules would act when coated on a fabric. Indeed, this appears to be the case as when the fabric is tested with both the dendrimer alone (ACR) and the complexed ACR the antibacterial results are virtually identical (similar to the disc diffusion assay). In addition, the cytotoxicity testing displays similar readings between the dendrimer alone and the complexed dendrimer meaning that I still am unable to understand why the complexed system is presented.

The results in the disc diffusion assay and in solution clearly showed that the complexed dendrimer is more effective than the ligand; MIC values of the complexed dendrimer were lower than those of the ligand at all the tested strains. Although virtually identical reduction of the antibacterial growth in the medium in presence of both treated cotton fabrics, the complexed dendrimer induces higher hydrophobicity of the cotton surface than the ACR ligand resulting in the absence of biofilm formation (Fig. 6C). This is the main reason for the synthesis and testing of the copper complex.

Round 2

Reviewer 2 Report

The authors have used a highly advanced 1H (600.13 MHz) and 13C (150.92
122 MHz) NMR spectrometer, I do not understand why they do not take advantage of such technique in order to finely characterize the new product.

Moreover, I can see that FTIR, EPR and other techniques have been well explained in the text and spectra are given in figures. Only detailed NMR spectra and data are not seen, which are considered as very important to characterize a new compound

Unfortunately, without a fine NMR characterization (1D and 2D spectra and data discussion) I cannot recommend the paper for publication

Good luck

Author Response

We accept any constructive criticism aimed at improving the quality of the proposed manuscript, but unfortunately in both reviews, the reviewer did not express her opinion on the presented  study.

We think the authors have the right to choose the type to present the characterisation data  - graphical , tabular or  textual. In this work, we have chosen to present this data as textual material in the experimental part, where the primary purpose is to characterize the structure of the compounds as is generally accepted in the scientific journals, including in "Materials". The right of the reviewer is to express his or her opinion essentially on the reliability of the evidence, but not to advise the authors on how to conduct and present their results. The graphical presentation in this study has been used for the copper complexes to show the difference between the dendrimer ligand and the corresponding copper complex.

I do not find the reviewer's attitude expressed in the sentence "Good luck" as serious and collegial.

Reviewer 3 Report

Upon review, the authors have amended the document appropriately and I would recommend publication in its current form.

Author Response

I thank the reviewer for the positive review.